# Interpolation and Regularization for Causal Learning

**Leena Chennuru Vankadara***
University of Tübingen, Tübingen AI Center

**Luca Rendsburg***
University of Tübingen, Tübingen AI Center

**Ulrike von Luxburg**
University of Tübingen, Tübingen AI Center

**Debarghya Ghoshdastidar**
Technical University of Munich

## Abstract

Recent work shows that in complex model classes, interpolators can achieve statistical generalization and even be optimal for statistical learning. However, despite increasing interest in learning models with good causal properties, there is no understanding of whether such interpolators can also achieve *causal generalization*. To address this gap, we study causal learning from observational data through the lens of interpolation and its counterpart—regularization. Under a simple linear causal model, we derive precise asymptotics for the causal risk of the min-norm interpolator and ridge regressors in the high-dimensional regime. We find a large range of behavior that can be precisely characterized by a new measure of *confounding strength*. When confounding strength is positive, which holds under independent causal mechanisms—a standard assumption in causal learning—we find that interpolators cannot be optimal. Indeed, causal learning requires stronger regularization than statistical learning. Beyond this assumption, when confounding is negative, we observe a phenomenon of self-induced regularization due to positive alignment between statistical and causal signals. Here, causal learning requires weaker regularization than statistical learning, interpolators can be optimal, and optimal regularization can even be negative.

## 1 Introduction

We consider the problem of learning the causal influence of multivariate covariates $x \in \mathbb{R}^d$ on a scalar target variable $y \in \mathbb{R}$ purely from observational data and under the presence of hidden confounders. Formally, given finite samples $\{(x_i, y_i)\}_{i=1}^n$ drawn independently and identically (i.i.d) from the joint *observational distribution* $p(x, y) = p(x)p(y|x)$, the goal of causal learning is to predict the effects on the target variable $y$ under interventions on the covariates $x$. In other words, the goal is to learn a predictive model that minimizes the expected loss on a random draw from the *interventional distribution* $p_{do}(x, y) = p(x)p(y|do(x))$, which can be different from the observational distribution.

Recently, Janzing (2019) established a close analogy between statistical and causal learning (albeit under a highly constructed confounded model). As a consequence, Janzing (2019) suggested that standard statistical learning-theoretic techniques (such as norm-based regularization) may also help learn good causal models. However, the classical statistical principles of bias-variance trade-off have been challenged in recent years by highly complex classes of models that are trained to interpolate the data and yet achieve remarkable generalization properties across a broad range of problem domains (Zhang et al., 2021). A large volume of recent work suggests that interpolation can be compatible with and may even be necessary to achieve optimal statistical generalization in the high-dimensional regime (Belkin et al., 2018; Belkin et al., 2019a; Liang et al., 2020; Feldman, 2020). Despite the surge in interest, causal properties of such interpolators have not yet been explored. In this work, we

---

* denotes equal contribution.

36th Conference on Neural Information Processing Systems (NeurIPS 2022).

consider a simple linear causal model in the high-dimensional regime ($n, d \to \infty, d/n \in \mathcal{O}(1)$) and ask: can interpolators achieve good causal generalization?

## 1.1 Motivation and Related Work

**Resemblance between statistical and causal generalization**    Causal learning can be regarded as an instance of the general problem of learning under distribution shifts, where the training (observational) distribution is shifted from the test (interventional) distribution. In the framework of out-of-distribution generalization, an interesting proposition for causal learning arises from the following high-level idea. Observing small sample sizes may induce a similar bias as distribution shifts. Therefore, techniques for learning models with good *out-of-sample* generalization (such as regularization) may also help to learn models with good *out-of-distribution* generalization and vice-versa. The literature provides plentiful evidence to support this general principle for different classes of distribution shifts. For instance, under a broad class of distribution shifts, distributionally robust optimization is equivalent to norm-based regularization (Xu et al., 2009; Shafieezadeh Abadeh et al., 2015; Gao et al., 2017; Shafieezadeh-Abadeh et al., 2019; Blanchet et al., 2019; Kuhn et al., 2019). Analogously, distributionally robust optimization techniques are also employed for statistical learning under limited samples (Zhu et al., 2020). Particularly relevant to our work is Janzing (2019), which formally establishes a close analogy between "generalizing from *empirical to observational distributions*" and "generalizing from *observational to interventional distributions*" under a highly constructed confounding model. This analogy suggests that standard norm-based regularization such as lasso or ridge, typically used for statistical learning, may also help learn better causal models.

**Interpolation can be compatible with statistical learning**    Explicit norm-based regularization techniques were initially motivated by the classical learning theory principle of bias-variance trade-off, which is characterized by a U-shaped generalization curve. This principle recommends to avoid interpolation and instead suggests to balance data fitting with the complexity of the hypothesis class. Recently, however, these classical principles have been challenged by deep learning models. Despite being highly complex with the ability to fit even random labels and often trained to interpolate the training data, they achieve state-of-the-art out-of-sample generalization across many domains (Zhang et al., 2021). A partial explanation is provided by the *double-descent* phenomenon (Belkin et al., 2019b; Belkin, 2021). Extending the generalization curve beyond the interpolation threshold reveals two regimes: the classical U-curve in the *underparameterized* regime and a decreasing curve in the *overparameterized* regime. This behaviour is not limited to deep neural networks, but extends to other settings such as random feature models and random forests (Belkin et al., 2019b; Hastie et al., 2022; Mei et al., 2021). Follow-up work suggests that in the overparameterized regime, interpolators can indeed achieve low statistical risk (Belkin et al., 2019a; Liang et al., 2020; Bartlett et al., 2020; Tsigler et al., 2020; Muthukumar et al., 2020).

**Is interpolation compatible with causal learning?**    On account of the parallels between statistical (out-of-sample) and causal (out-of-distribution) learning, it is therefore natural to ask: *can interpolators also learn good causal models?* One line of empirical work suggests that naively applying distributionally robust learning techniques such as importance reweighting or distributionally robust optimization approaches (which are equivalent to certain forms of regularization) offers vanishing benefits over empirical risk minimization in overparameterized model classes (Byrd et al., 2019; Sagawa et al., 2020; Gulrajani et al., 2021). However, there is also empirical evidence that suggests that augmenting such techniques with additional explicit norm-based regularization may help to learn distributionally robust models in the overparameterized regime (Sagawa et al., 2020; Donhauser et al., 2021). In the context of causal learning, it has been suggested that explicit regularization can be beneficial and might even need to be stronger than for statistical learning (Janzing, 2019; Vankadara et al., 2021). Existing work, therefore, remains unclear about the role of explicit regularization in causal learning, or correspondingly, whether interpolation is compatible with causal learning. In this work, we take a theoretical approach to systematically address these questions.

## 1.2 Our Contributions

We provide a first analysis of causal generalization from observational data in the modern, overparameterized and interpolating regime under a simple linear causal model. Specifically, we consider the interpolating minimum $l_2$ norm least-squares estimator and the family of regularized ridge regression

estimators in the proportional asymptotic regime. We seek answers to the following questions: is there a regime where the optimal causal regularization parameter is 0, that is, can we observe *benign causal overfitting*? Furthermore, if the optimal causal regularization parameter is positive, how strongly do we need to regularize? How does the optimal causal regularization compare to the optimal statistical regularization? While our analysis is exhaustive, we emphasize the results under the assumption of independent causal mechanisms (Janzing et al., 2010), a standard assumption in causal learning.

- **Precise asymptotics of the causal risk (Section 3).** We provide precise asymptotics of the *causal risk* of the ridge regression estimator as well as the minimum $l_2$ norm interpolator in the high-dimensional setting: $n, d \to \infty, d/n \to \gamma \in (0, \infty)$. Our results confirm that, similar to the statistical setting, the causal generalization curve of the min-norm estimator exhibits the double-descent phenomenon. This is because the variance term diverges at the interpolation threshold and is decreasing in the overparameterized regime ($\gamma > 1$).

- **A measure of confounding strength $\zeta$ (Section 2.1).** We introduce a new measure of *confounding strength* $\zeta$ that quantifies the relative contribution of the "confounding signal" to the "causal signal". It can be interpreted as the strength of the distribution shift between the observational and interventional distributions. While $\zeta$ can take any real value in general, it is restricted to $[0, 1]$ under the assumption of independent causal mechanisms. There, it induces a strict, model-independent ordering of all causal models that entail the same observational distribution.

- **Benign causal overfitting (Section 4).** When the causal signal dominates the statistical signal ($\zeta < 0$), we observe a phenomenon of self-induced regularization due to the confounding signal. As a consequence, the optimal causal regularization can be 0 or negative even if the optimal statistical regularization is strictly positive. Under the assumption of independent causal mechanisms, however, we show that there is no benign causal overfitting. This is in contrast to benign statistical overfitting, which can occur in the highly underparameterized regime ($\gamma \to 0$).

- **Optimal causal vs. statistical regularization (Section 5).** We show that causal learning requires weaker regularization than statistical learning when the confounding strength $\zeta$ is negative. However, when $\zeta > 0$ and in particular under the principle of independent causal mechanisms, we show that causal learning requires stronger regularization than statistical learning. More specifically, the optimal causal regularization is strictly increasing in confounding strength.

## 2 Problem Setup

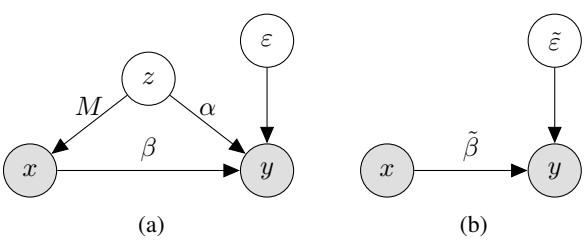

Figure 1: (*a*) Graphical model of the causal model defined in (1). (*b*) The usual statistical model. In both figures, observed random variables are shaded and unobserved variables are white.

We consider a linear causal model with parameters $M \in \mathbb{R}^{d \times l}, \alpha \in \mathbb{R}^l, \beta \in \mathbb{R}^d$ with $l \geq d$ and $\sigma^2 > 0$ described via the *structural equations*

$$z \sim \mathcal{N}(0, I_l), \quad \varepsilon \sim \mathcal{N}(0, \sigma^2), \quad x = Mz, \quad y = x^T \beta + z^T \alpha + \varepsilon. \tag{1}$$

The covariates $x \in \mathbb{R}^d$ and the target $y \in \mathbb{R}$ are *confounded* through $z \in \mathbb{R}^l$, which follows a standard normal distribution. This structure implies that $\mathbb{E}x = 0$ and the covariance of $x$ is $\Sigma := \text{Cov}\, x = MM^T$. A graphical representation of this causal model is given in Figure 1a. The observational joint distribution of this causal model is given by $p(x, y) = p(x)p(y|x)$, where $x \sim \mathcal{N}(0, \Sigma)$ and $y|x \sim \mathcal{N}(x^T \tilde{\beta}, \tilde{\sigma}^2)$. Here, the statistical parameter $\tilde{\beta} := \beta + \Gamma$ consists of the causal parameter $\beta$ and a confounding parameter $\Gamma := M^{+T}\alpha$, where $M^{+T}$ is shorthand for $(M^+)^T$ and $M^+$ denotes the Moore-Penrose inverse of $M$. The statistical noise is given by $\tilde{\sigma}^2 := \sigma^2 + \|\alpha\|^2 - \|\Gamma\|_\Sigma^2$, where $\|x\|_\Sigma^2 := x^T \Sigma x$ denotes the generalized norm. [1] Note that the observational distribution alone cannot distinguish the causal model from the one in Figure 1b. The

---

[1] Note that $\|\alpha\|^2 - \|\Gamma\|_\Sigma^2 = \|\alpha\|_{I-M^+M}^2 \geq 0$, where $I - M^+M$ is the orthogonal projection onto ker $M$.

goal of statistical learning is to predict $y$ after observing $x$, which is captured by the conditional distribution $p(y|x)$. In contrast, the goal of causal learning is to predict $y$ after manipulating or intervening on $x$. This is formally captured by Pearl's *do*-calculus (Pearl, 2009), which describes interventions on random variables as a shift in the joint distribution. Intervening on $x$ with the value $x_0$, denoted as $do(x = x_0)$, removes all arrows to $x$ and *sets* $x = x_0$. In our causal model (1), the intervention $do(x = x_0)$ removes the arrow $z \to x$ and yields the updated structural causal equations

$$z \sim \mathcal{N}(0, I_l), \quad \varepsilon \sim \mathcal{N}(0, \sigma^2), \quad x = x_0, \quad y = x_0^T \beta + z^T \alpha + \varepsilon.$$

The corresponding distribution of $y$ after intervening on $x$ is therefore given by $y|do(x = x_0) \sim \mathcal{N}(x_0^T \beta, \tilde{\sigma}^2 + \|\Gamma\|_\Sigma^2)$. Since arbitrary interventions can introduce arbitrary distribution shifts, we consider the natural class of interventions drawn from the observational marginal distribution on $x$. This yields the interventional joint distribution $p_{do}(x, y) = p(x)p(y|do(x))$ with the slight abuse of notation $do(x)$ in which the random variable $x$ and its value coincide.

**Causal learning from observational data** Assume we are given i.i.d. samples $\{(x_i, y_i)\}_{i=1}^n$ from the observational joint distribution $p(x, y)$, which we collect in $X \in \mathbb{R}^{n \times d}$ and $Y \in \mathbb{R}^n$. The usual statistical learning aims for the observational conditional $p(y|x)$, which means that train and test distributions coincide. Causal learning aims for the interventional conditional $p(y|do(x))$, a distribution shift problem for which train and test distributions differ. We define the corresponding *causal risk* $R^C$ and *statistical risk* $R^S$ of any linear regressor $\hat{\beta} \in \mathbb{R}^d$ under the squared loss as

$$R^C(\hat{\beta}) := \mathbb{E}_x \mathbb{E}_{y|do(x)} (x^T \hat{\beta} - y)^2 \quad \text{and} \quad R^S(\hat{\beta}) := \mathbb{E}_x \mathbb{E}_{y|x} (x^T \hat{\beta} - y)^2. \tag{2}$$

The following proposition (proven in Appendix A) characterizes the risks under the model (1).

**Proposition 2.1 (Causal and Statistical Risk).** *For any $\hat{\beta} \in \mathbb{R}^d$, the risks defined in Eq. (2) satisfy*

$$R^C(\hat{\beta}) = \|\hat{\beta} - \beta\|_\Sigma^2 + \tilde{\sigma}^2 + \|\Gamma\|_\Sigma^2 \quad \text{and} \quad R^S(\hat{\beta}) = \|\hat{\beta} - \tilde{\beta}\|_\Sigma^2 + \tilde{\sigma}^2.$$

Therefore, $\beta$ is the optimal causal parameter and $\tilde{\beta}$ is the optimal statistical parameter. In the following, we simply refer to them as causal and statistical parameters.

## 2.1 A New Measure of Confounding Strength

Since the interventional distribution generally differs from the observational distribution, we require a measure that quantifies how this shift influences causal learning from observational data.

**Signal-to-noise ratios (SNRs)** Before we define our measure of confounding strength, we first define the statistical and causal signal-to-noise ratios, which help to intuitively understand our confounding strength measure. Recall that every causal model entails a statistical model since the causal parameter $\beta$ and the confounding parameter $\Gamma$ jointly specify the statistical parameter $\tilde{\beta} = \beta + \Gamma$. The statistical SNR is defined as usual by $\text{SNR}_S := \|\tilde{\beta}\|^2 / \tilde{\sigma}^2$. For the causal SNR, a natural notion would be $\|\beta\|^2 / (\tilde{\sigma}^2 + \|\Gamma\|_\Sigma^2)$ if the learning algorithm had access to data from the interventional distribution $y|do(x) \sim \mathcal{N}(x^T \beta, \tilde{\sigma}^2 + \|\Gamma\|_\Sigma^2)$; but since we are constrained to data from the observational conditional $y|x \sim \mathcal{N}(x^T \tilde{\beta}, \tilde{\sigma}^2)$, the corresponding causal SNR, which quantifies the hardness of the learning problem, needs to take this into consideration. Accordingly, we consider the causal SNR as the ratio of the alignment between the statistical and causal parameters and the variance of the observational conditional. Formally, we define it as $\text{SNR}_C := \langle \beta, \tilde{\beta} \rangle / \tilde{\sigma}^2$. In what follows, we therefore often refer to $\langle \beta, \tilde{\beta} \rangle$ as the *causal signal* and $\|\tilde{\beta}\|^2$ as the *statistical signal*. Correspondingly, we refer to $\langle \tilde{\beta} - \beta, \tilde{\beta} \rangle = \langle \Gamma, \tilde{\beta} \rangle$ as the *confounding signal*, which is the alignment between the confounding parameter $\Gamma$ and the statistical parameter $\tilde{\beta}$.

**Confounding strength** Regression on observational data implicitly assumes that the interventional distribution coincides with the observational distribution, while it can be shifted in general. To quantify the impact of this distribution shift on the corresponding causal risk, we introduce a new *confounding strength measure* $\zeta$. It measures the relative contribution of the confounding signal to the statistical signal and is defined by

$$\zeta := \frac{\langle \Gamma, \tilde{\beta} \rangle}{\langle \Gamma, \tilde{\beta} \rangle + \langle \beta, \tilde{\beta} \rangle} = \frac{\langle \Gamma, \tilde{\beta} \rangle}{\|\tilde{\beta}\|^2}. \tag{3}$$

Other notions of confounding strength are possible, but we will see later that this definition is well-suited to capture the shift strength for causal learning from observational data. Without further restrictions, $\zeta$ can take any value in $\mathbb{R}$. This measure divides the causal models into the following three regimes, depending on the relationship between causal and statistical signal:

- $\zeta \geq 1$: the causal signal $\langle \beta, \tilde{\beta} \rangle$ is non-positive, which implies that causal and statistical parameters are orthogonal or negatively aligned. Statistical learning is adversarial to causal learning.
- $0 < \zeta < 1$: causal and statistical parameters are positively aligned but the causal signal is weaker than the statistical signal $\|\tilde{\beta}\|^2$, for example $\beta = \tilde{\beta}/2$.
- $\zeta \leq 0$: the causal signal dominates the statistical signal, for example $\beta = 2\tilde{\beta}$.

The SNRs are related to the confounding strength measure via $\mathrm{SNR_C} = (1 - \zeta)\,\mathrm{SNR_S}$. In particular, the causal signal decreases as the confounding strength increases.

**The regime $0 \leq \zeta \leq 1$ is practically most relevant** Causal learning often requires strong assumptions because causal models cannot be uniquely identified by their observational distribution. A standard assumption is the principle of independent causal mechanisms (ICM) (Janzing et al., 2010; Lemeire et al., 2013; Peters et al., 2017), which informally asserts that causal mechanisms share no information. In our causal model (1), a corresponding assumption could be that the causal mechanisms $\alpha$ and $\beta$ are drawn from rotationally invariant distributions. This implies that $\langle \beta, \Gamma \rangle \to 0$ as $d \to \infty$, which in turn falls in the regime $0 \leq \zeta \leq 1$. While our following analysis covers all possible causal models, we pay special attention to this regime because it might be of highest practical relevance. Note that for $\langle \beta, \Gamma \rangle = 0$, our measure of confounding strength coincides with the measure $\zeta' = \|\Gamma\|^2 / (\|\Gamma\|^2 + \|\beta\|^2)$ introduced by Janzing et al. (2017). It measures the relative contribution of causal and confounding signal in terms of lengths rather than inner products.

## 3 Causal and Statistical Risk of High-Dimensional Regression Models

Causal learning is extremely challenging, because it requires scarcely available interventional data or has to rely on other information such as exogenous (Rothenhäusler et al., 2021) or instrumental variables (Angrist et al., 1991). In our setting where only observational data are available, causal learning requires additional model assumptions. One such approach has been followed by the Concorr method (Janzing, 2019) which leverages the ICM assumption to make an improved choice of regularization parameter under a linear regression model. To fully characterize the effect of regularization on causal generalization, we consider two estimators for learning causal models from observational data $(X, Y) \in \mathbb{R}^{n \times d} \times \mathbb{R}^n$: the min-norm interpolator and ridge regressors. The *min-norm interpolator* is the minimum $l_2$ norm solution to the least squares regression problem

$$\hat{\beta}_0(X, Y) \coloneqq \arg\min\{\|\hat{\beta}\|_2 : \hat{\beta} \in \arg\min_{\hat{\beta} \in \mathbb{R}^d} \|Y - X\hat{\beta}\|^2\}. \tag{4}$$

A closed form is given by $\hat{\beta}_0(X, Y) = (X^T X)^+ X^T Y$. For $\lambda > 0$, the *ridge regressor* solves

$$\hat{\beta}_\lambda(X, Y) \coloneqq \arg\min_{\hat{\beta} \in \mathbb{R}^d} \frac{1}{n}\|Y - X\hat{\beta}\|^2 + \lambda\|\hat{\beta}\|^2\,, \tag{5}$$

which has the explicit solution $\hat{\beta}_\lambda(X, Y) = (X^T X + n\lambda I_d)^{-1} X^T Y$. The min-norm interpolator can be obtained as a limiting case from the ridge regression solution via $\hat{\beta}_0(X, Y) = \lim_{\lambda \to 0^+} \hat{\beta}_\lambda(X, Y)$. Whenever it is clear from the context, we drop the dependence of the predictors on $X$ and $Y$.

Before proceeding with the analysis, we motivate the idea that appropriate regularization can help to learn causal models from purely observational data. To this end, we compare regularization chosen by statistical cross validation to regularization based on an *interventional validation set* in Figure 2. Since cross validation implicitly assumes that there is no confounding, it is close to Bayes optimal for $\zeta = 0$ when $n \gg d$. However, as confounding increases, it falls behind regularization based on the interventional validation set. The latter even yields Bayes optimal risk again in the purely confounded setting $\zeta = 1$, where the lack of causal signal ($\beta = 0$) is encoded by infinite regularization. While we might not have access to an interventional validation set in practice, our theory will show that knowledge of confounding strength is sufficient for choosing appropriate regularization. Finally, we want to caution that even though regularization can help, it does not remove the hardness of causal learning. Reliable causal inference still requires stronger assumptions or additional data.

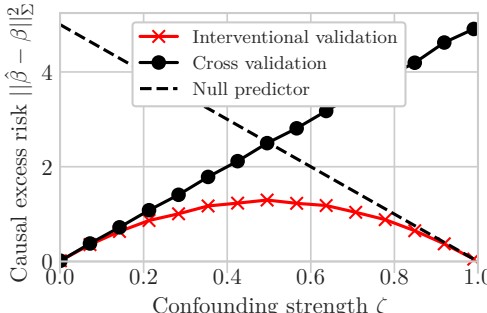

Figure 2: Causal excess risk of ridge predictors based on $n = 30,000$ samples from the observational distribution. Regularization is chosen either by cross validation or based on a validation set from the interventional distribution of same size. Each model has fixed dimensions $d = 300, l = 350$ and $\text{SNR}_\text{S} = 5$, but different underlying confounding strengths under the constraint $\langle \beta, \Gamma \rangle = 0$. The benefits of optimal regularization over cross validation increase with confounding strength.

## 3.1 Precise Asymptotics of the Causal and Statistical Risks

In this section, we provide precise asymptotics for the causal and statistical risks of the min-norm interpolator and ridge regression solutions in the high-dimensional regime. This regime is characterized by both $n, d \to \infty$ such that $d/n \to \gamma \in (0, \infty)$, where $\gamma$ is called the *overparameterization ratio*. We distinguish between the *underparameterized regime* ($\gamma < 1$) and the *overparameterized regime* ($\gamma > 1$). All proofs for this section are deferred to Appendix B. Since the predictors $\hat\beta = \hat\beta(X, Y)$ are random variables in the training data $X$ and $Y$, so is their corresponding causal risk. We consider the expectation of this risk under $Y$ conditioned on $X$. According to Proposition 2.1, it is given by $R_X^C(\hat\beta) := \mathbb{E}_{Y|X} R^C(\hat\beta) = \mathbb{E}_{Y|X} \|\hat\beta - \beta\|_\Sigma^2 + \tilde\sigma^2 + \|\Gamma\|_\Sigma^2$. Due to its simple form, similar to the usual statistical risk, the causal excess risk can be decomposed into bias and variance:

$$\mathbb{E}_{Y|X}\|\hat\beta - \beta\|_\Sigma^2 = \underbrace{\|\mathbb{E}_{Y|X}\hat\beta_\lambda - \beta\|_\Sigma^2}_{=:B_X^C(\hat\beta_\lambda)} + \underbrace{\mathbb{E}_{Y|X}\|\hat\beta_\lambda - \mathbb{E}_{Y|X}\hat\beta_\lambda\|_\Sigma^2}_{=:V_X^C(\hat\beta_\lambda)} . \tag{6}$$

The next theorem is one of our main results. It gives a closed-form expression for the limiting causal bias and variance of the min-norm interpolator and ridge regression estimators. We make the simplifying assumption of isotropic covariance $\Sigma = I_d$. The proof relies on recent techniques from random matrix theory. It employs arguments similar to Dicker (2016), Dobriban et al. (2018), and Hastie et al. (2022) and can correspondingly be extended to arbitrary covariances under boundedness assumptions on the spectrum. We leave such extensions for future work and focus on thoroughly understanding the isotropic causal model, because it already exhibits rather rich behavior.

**Theorem 3.1 (Limiting Causal Bias-Variance Decomposition for the Ridge Estimator).** *Let* $\|\beta\|^2 = r^2$, $\|\Gamma\|^2 = \omega^2$, $\langle \Gamma, \beta \rangle = \eta$, *and fix* $\tilde\sigma^2$. *Then as* $n, d \to \infty$ *such that* $d/n \to \gamma \in (0, \infty)$, *it holds almost surely in* $X$ *for every* $\lambda > 0$ *that*

$$B_X^C(\hat\beta_\lambda) \to \mathcal{B}_\lambda^C = \omega^2 + \tilde{r}^2 \lambda^2 m'(-\lambda) - 2(\omega^2 + \eta)\lambda m(-\lambda) \quad and \tag{7}$$

$$V_X^C(\hat\beta_\lambda) \to \mathcal{V}_\lambda^C = \tilde\sigma^2 \gamma(m(-\lambda) - \lambda m'(-\lambda)), \tag{8}$$

*where* $m(\lambda) = ((1 - \gamma - \lambda) - \sqrt{(1 - \gamma - \lambda)^2 - 4\gamma\lambda})/(2\gamma\lambda)$ *and* $\tilde{r}^2 = r^2 + \omega^2 + 2\eta$. *Therefore* $R_X^C(\hat\beta_\lambda) \to \mathcal{R}_\lambda^C = \mathcal{B}_\lambda^C + \mathcal{V}_\lambda^C + \tilde\sigma^2 + \omega^2$. *The corresponding limiting quantities for the min-norm interpolator can be obtained by taking the limit* $\lambda \to 0^+$ *in* (7) *and* (8).

From these limiting expressions we can see that the causal risk curve of the min-norm interpolator exhibits the double descent phenomenon: it diverges at the interpolation threshold $\gamma = 1$ due to the variance term and decreases again for $\gamma > 1$. A corresponding visualization is given in Figure 4. Explicit regularization dampens the divergence of the variance term. While we are primarily interested in the causal risk, the corresponding statistical risk serves as a natural baseline. An analogue set of results for the statistical risk is given in Appendix C. These results have already been derived by Hastie et al. (2022) and can also be recovered as a special case of our causal results: for fixed statistical parameters $\tilde\beta$ and $\tilde\sigma^2$, the statistical risk coincides with the causal risk of an unconfounded causal model defined with $\beta = \tilde\beta$, $\sigma^2 = \tilde\sigma^2$, and $\alpha = 0$. In particular, the corresponding statistical limiting expressions are the same as in Theorem 3.1 after setting $\eta = \omega^2 = 0$.

**Optimal statistical and causal regularization** By directly optimizing the closed form expressions for limiting causal and statistical risks we can find the optimal causal and statistical regularization.

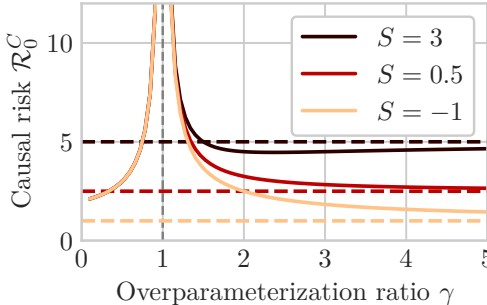
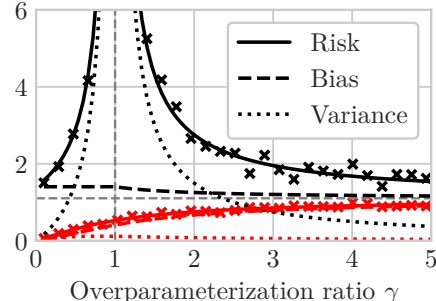

Figure 3: Limiting causal excess risk $\mathcal{R}_0^C$ (without the constant $\tilde{\sigma}^2 + \omega^2$) of the min-norm interpolator for different causal signal strengths $S$. Dashed lines are the corresponding null-risks $\omega^2$, which are outperformed more often as $S$ increases. For $\gamma < 1$, all three curves coincide.

Figure 4: Limiting bias-variance decomposition and causal excess risk of the min-norm interpolator (black) and optimally regularized ridge regression (red). Crosses indicate finite-sample risks of $n = d/\gamma$ samples with $d = 300$. The finite risks are well-predicted by their theoretical limit.

For any $\gamma \in (0, \infty)$, the optimal statistical regularization $\lambda_S^*(\gamma) := \arg\inf_{\lambda \in (0, \infty)} \mathcal{R}_\lambda^S$ can be expressed in closed-form as $\lambda_S^*(\gamma) = \mathrm{SNR_S}^{-1} \gamma$. The closed-form expression for the optimal causal regularization parameter $\lambda_C^*(\gamma) := \arg\inf_{\lambda \in (0, \infty)} \mathcal{R}_\lambda^C$ is a root of a 4th order polynomial and as such considerably intricate. For readability, we do not include it here. We investigate the behavior of the optimal causal and statistical regularization in Section 4 and 5.

### 3.2 Basic Behavior of the Limiting Risk

We start to analyze the results by assessing the basic behavior of the limiting causal risk. The causal risk of the null estimator $\hat{\beta} = 0$ serves as a natural baseline to evaluate the performance of the the min-norm interpolator and the ridge regression estimators.

**Regimes of the min-norm interpolator** Theorem 3.1 characterizes the limiting causal risk of the min-norm interpolator. Its behavior is controlled by the causal signal-to-noise ratio, which we defined as $\mathrm{SNR_C} = (1 - \zeta)\,\mathrm{SNR_S}$. However, as we will later see, the causal risk of the min-norm interpolator can be lower than null risk when $\zeta < 0.5$. To distinguish the regimes of the min-norm interpolator, it is therefore convenient to consider the closely related quantity $S = (1 - 2\zeta)\,\mathrm{SNR_S}$. It distinguishes between three different regimes (visualized in Figure 3).

- For $S > 1$, the causal signal dominates the noise and the min-norm interpolator can perform better than null risk in both under- and overparameterized regime.

- For $0 \leq S \leq 1$, the causal signal is weaker than the noise. Only the underparameterized regime can beat the null risk, whereas the overparameterized regime is always worse.

- The previous two cases resemble the behavior of the statistical risk in the corresponding regimes of the statistical SNR. Contrary to the statistical risk, however, the causal risk admits a third regime $S < 0$. In this case, the min-norm interpolator always performs worse than null risk. Here, the causal signal $\langle \beta, \tilde{\beta} \rangle$ is dominated by the confounding signal $\langle \Gamma, \tilde{\beta} \rangle$, and interpolating the observational data overfits to the confounding.

**Bias and variance** The bias-variance decomposition of the causal risk given in Theorem 3.1 is visualized in Figure 4 for the min-norm interpolator and the optimally ridge-regularized regressor. The figure also shows the causal risk based on finite samples from the model, which is in high agreement with our asymptotic results. We compare the causal risk to the corresponding statistical risk. First note that the causal and statistical variance terms coincide exactly for both the min-norm interpolator and ridge regressors. This is because the variance term of the squared loss depends only on the variance in the training data, but not on the target parameter $\beta$ or $\tilde{\beta}$. Since the training data are the same for both causal and statistical learning, the variance terms trivially coincide.

For the min-norm interpolator, as in the statistical case, the variance term is responsible for the double-descent behavior of the causal risk curve because it explodes at the interpolation threshold $\gamma = 1$ and decreases in the overparameterized regime $\gamma > 1$. In the statistical setting, the bias strictly increases in the overparameterized regime and, as a consequence, the best risk is always achieved in the underparameterized setting. In contrast, the causal bias of the min-norm interpolator can be decreasing in the overparameterized regime and therefore the optimal causal risk can be achieved in the highly overparameterized regime $\gamma \to \infty$. However, this only happens in the regime $S < 0$ where the risk of the min-norm interpolator is always worse than null risk.

Figure 4 shows the causal risk of the optimally regularized ridge regression estimator which trivially is always below that of the min-norm risk. Similar to the statistical setting, the corresponding generalization curve does not exhibit the double descent phenomenon. There are qualitatively different reasons for why regularization helps in statistical and causal learning. For both statistical and causal learning, regularization decreases the shared variance, which corresponds to the finite-sample error. However, while the statistical bias always increases with regularization, the causal bias can actually decrease. This implies that regularization not only helps with the finite-sample error, but can also reduce the error due to confounding.

**Higher confounding implies higher causal risk for all $\lambda$**    So far, we have investigated the causal risk under a single causal model. Now we can compare different causal models using the confounding strength measure $\zeta$ introduced in Section 2.1. The next proposition shows that $\zeta$ governs the hardness of causal learning from observational data. Specifically, the causal risk of the ridge regression for any $\lambda \in (0, \infty)$ increases as the causal model becomes more confounded. A proof is given in Appendix D.

**Proposition 3.2 (Causal Risk Increases with Confounding Strength).** *Consider the family of causal models parameterized as in (1) that entail the same observational distribution. Let $C_1$ and $C_2$ be two such causal models with confounding strengths $\zeta_1$ and $\zeta_2$ and alignments $\eta_1$ and $\eta_2$ (defined in Theorem 3.1), respectively. Then for all $\lambda, \gamma \in (0, \infty)$,*

$$\zeta_1 > \zeta_2, \ \ \eta_1 \leq \eta_2 \implies \mathcal{R}_\lambda^{C_1} > \mathcal{R}_\lambda^{C_2}.$$

*In particular, for any fixed $\eta$, the measure of confounding strength $\zeta$ establishes a strict ordering of causal models. This includes the ICM under which $\eta = 0$.*

## 4    Benign Causal Overfitting

A large number of recent works suggest that minimum-norm interpolators can be optimal for statistical generalization (Belkin et al., 2018; Belkin et al., 2019a; Muthukumar et al., 2020). This phenomenon is often referred to as benign overfitting. Moreover, the optimal statistical generalization may even be achieved for negative regularization $\lambda < 0$ (Kobak et al., 2020; Bartlett et al., 2020; Tsigler et al., 2020). It is unclear, however, if such interpolators, which have implicit small-norm biases, can also be optimal when there is a shift between the training and test distributions. In particular, we ask: can the optimal causal regularization be 0 or even negative, that is, do we observe *benign causal overfitting?* To show that the optimal regularization can be negative, we simply show that the derivative of the causal risk at 0 is positive. We summarize our key findings in Theorem 4.1.

**Theorem 4.1 (Optimal Regularization can be Negative).** *For any causal model parameterized as in (1), the following cases distinguish between whether the min-norm interpolator is optimal or not.*

1. *For negative confounding strength $\zeta < 0$ the optimal causal regularization $\lambda_C^*$ can be 0 or even negative. A necessary and sufficient condition for $\lambda_C^* \leq 0$ depends on the difference in causal and statistical signal-to-noise ratios and is given by*

$$\mathrm{SNR_C} - \mathrm{SNR_S} \geq \frac{\gamma \max\{1, \gamma\}}{(1 - \gamma)^2}.$$

2. *For positive confounding strength $\zeta > 0$ the optimal causal regularization is positive $\lambda_C^* > 0$ and $\mathcal{R}_0^C > \mathcal{R}_{\lambda_C^*}^C$, hence regularization is beneficial. This includes the ICM.*

In the highly overparameterized regime ($\gamma \to \infty$), the benefit of explicit regularization vanishes and both the causal and statistical risks of the ridge regression estimator converge to their corresponding

null risks, independently of the regularization. We do not refer to this as benign overfitting. However, we can observe benign causal overfitting when the causal SNR is larger than the statistical SNR ($\zeta < 0$), which happens when causal and statistical parameter are strongly aligned. This implies that the norm of the statistical parameter is smaller than the norm of the causal parameter. Consequentially, statistical regressors are implicitly biased towards solutions of smaller norm and causal learning exhibits self-induced regularization. Compare this to benign statistical overfitting, which happens for certain alignments between the regression parameter $\tilde{\beta}$ and the covariance matrix $\Sigma$. In our isotropic setting $\Sigma = I_d$, we can therefore never observe benign statistical overfitting, but we can observe benign causal overfitting. This phenomenon occurs in both the underparameterized as well as the overparameterized regime. The range of $\gamma$ for which the optimal causal regularization is negative increases with the dominance of the causal signal over the statistical signal. As $\gamma$ approaches the interpolation threshold, it becomes harder for the optimal causal regularization to be negative. When the causal SNR is smaller than the statistical SNR ($\zeta > 0$) and in particular under the ICM ($0 < \zeta \leq 1$), the optimal causal regularization is strictly positive and the benefit of explicit regularization does not vanish. This can be the case even when the optimal statistical regularization vanishes. To see this consider the statistical risk in the highly underparameterized regime $\gamma \to 0$. In this regime, the benefit of explicit regularization vanishes and the min-norm interpolator indeed achieves the optimal *statistical* risk. The optimal causal regularization is given explicitly by $\lambda_C^* = \zeta/(1 - \zeta)$ for $0 \leq \zeta \leq 1$ and $\lambda_C^* = \infty$ for $\zeta > 1$. This is strictly positive and increasing in the confounding strength $\zeta$, and in fact diverges as $\zeta$ approaches 1 (see Theorem 5.2).

# 5 On Optimal Regularization

In this section, we investigate two key questions which are natural in the context of our work. How does the optimal causal regularization $\lambda_C^*$ compare to the optimal statistical regularization $\lambda_S^*$? What is the dependence of the optimal causal regularization $\lambda_C^*$ on the confounding strength $\zeta$?

**Optimal statistical vs. causal regularization**   When the training and test distributions coincide, approaches such as cross-validation or information criteria (for example AIC or BIC) can be used to estimate the regularization parameter for optimal out-of-sample generalization. However, choosing the correct regularization parameter for causal learning can be challenging without interventional data. To understand the optimal causal regularization, it is natural to compare it to the optimal statistical regularization, which can usually be estimated from data. Interestingly, our analysis reveals that when confounding strength is positive $\zeta > 0$ and in particular under the ICM one needs to regularize more strongly for causal generalization than for statistical generalization. However, when the confounding strength is negative, that is, when the causal signal dominates the statistical signal, the optimal causal regularization $\lambda_C^*$ can actually be smaller than the optimal statistical regularization $\lambda_S^*$. We formally present this result in Theorem 5.1.

**Theorem 5.1 (Optimal Statistical vs. Causal Regularization).** *For any causal model parameterized as in (1), the condition $\zeta = 0$ defines a phase transition for the optimal regularization via*

$$\zeta < 0 \iff \lambda_C^* < \lambda_S^*, \qquad \zeta = 0 \iff \lambda_C^* = \lambda_S^*, \quad and \quad \zeta > 0 \iff \lambda_C^* > \lambda_S^*.$$

*In particular under the ICM, the optimal causal regularization $\lambda_C^*$ is always strictly larger than the optimal statistical regularization $\lambda_S^*$, unless $\zeta = 0$, in which case they coincide.*

**Dependence on confounding strength $\zeta$**   The problem of causal learning from observational data is a distribution shift problem where the distribution of the training data is shifted from that of the test distribution. As discussed earlier in Proposition 3.2, the confounding strength measure $\zeta$ quantifies the strength of this distribution shift. Therefore, we expect the optimal causal regularization to increase with confounding strength. Theorem 5.2 indeed confirms this intuition.

**Theorem 5.2 (Increasing Confounding Strength Requires Stronger Regularization).** *Consider the family of causal models parameterized as in (1) that entail the same observational distribution. The optimal causal regularization $\lambda_C^*$ only depends on the confounding strength $\zeta$ and $\lambda_C^*$ is an increasing function in $\zeta$. More specifically, using $\varrho = -\mathrm{SNR_S}^{-1}\gamma \max\{1, \gamma\}/(1 - \gamma)^2$:*

$$\varrho < \zeta < 1 \implies \lambda_C^* \in (0, \infty) \text{ with } \partial_\zeta \lambda_C^* > 0,$$

*$\lambda_C^* = 0$ if $\zeta \leq \varrho$ and $\lambda_C^* = \infty$ for $\zeta \geq 1$.*

# 6   Summary and Extensions

We characterize the role of explicit regularization for causal learning from observational data by computing the asymptotic risk of ridge-regularized regressors and the min-norm interpolator (Theorem 3.1). Under the principle of independent causal mechanisms (ICM), we find that causal learning requires stronger regularization than statistical learning (Theorem 5.1). A practical implication is that the regularization parameter for causal learning should be chosen larger than what is suggested by cross-validation. We can precisely state how much larger based on an estimate of confounding strength (Janzing et al., 2017; Janzing et al., 2018). Beyond ICM, we show that strong alignments between causal and statistical parameters can cause self-induced regularization and lead to benign causal overfitting (Theorem 4.1). One could consider generalizing our assumptions: arbitrary covariances, shifts in the marginal distributions of covariates, soft interventions, more complex hypothesis classes, or non-linear causal relationships. Since the linear model already exhibits rich behavior, we focus in this paper on understanding the simple setting. Below, we briefly discuss extensions of our analysis to causal learning under soft interventions, non-linearity, and non-Gaussianity.

**Soft interventions**    It is not always appropriate to consider causal learning under hard interventions. Instead, it is often of interest to consider *soft interventions*. In these settings, the qualitative statements derived from our analysis still hold. To illustrate this, we consider the class of shift interventions where the structural dependence of the covariates $x$ is not destroyed as in the case of hard interventions but the observed covariates are merely perturbed (i.e., interventions of the form $do(x := x + \nu)$). Then it turns out that Causal risk$_{\text{soft}}$ = Causal risk$_{\text{hard}}$ + Statistical risk. From our results, it then follows that under ICM, $\lambda^{\text{statistical}} \leq \lambda^{\text{causal}}_{\text{soft}} \leq \lambda^{\text{causal}}_{\text{hard}}$ This also supports our intuition since under soft interventions, we typically aim to achieve a tradeoff between statistical and causal predictability. We include a complete analysis under shift interventions in Appendix F.

**Extensions to non-linear models**    It is feasible to extend the analysis to structural causal models that arise in a reproducing kernel Hilbert space corresponding to a positive definite kernel (i.e, where the best statistical model $\tilde{f}$ and the best causal model $f$ are functions in some RKHS). There are two major technical challenges to deriving the theoretical analysis in such non-linear settings. Both are beyond what can be done in this paper and are left for future work, but we briefly outline them below.

1. **Extend the definition of confounding strength $\zeta$ beyond the linear setting.** Since such a definition is non-trivial already in the linear setting, it is challenging to meaningfully generalize this to the non-linear setting. However, under non-linear causal models in the RKHS, we can naturally extend this definition by replacing the Euclidean norms with functional norms in the RKHS. Generalizing the analysis beyond this setting would require further careful consideration.

2. **Derive limiting expressions for causal risk of regularized regressors in a non-linear hypothesis class.** In the case of kernel regression, this would still be feasible via recent random matrix theory results [27]. By optimizing the limiting expressions with respect to the regularization parameter, one can obtain the parameter that achieves the optimal causal risk and subsequently identify the relationship between optimal causal regularization and confounding strength.

**Beyond Gaussianity**    The analysis can be extended beyond the Gaussian setting by considering random variables generated by finite mixtures of Gaussians. Due to the universality phenomenon in the high-dimensional limit, we believe that our limiting expressions (and the qualitative messages derived henceforth) would be rather robust to shifts in the marginal distribution as long as moments of order $(4 + \delta)$ for some $\delta > 0$ are bounded. We conducted experiments to verify this claim and the corresponding results can be found in Appendix G. They show that for distributions with finite 4th moments, the finite-sample risks of the min-norm interpolator and causally optimally regularized ridge regressor closely match the theoretically derived asymptotic risks.

## Acknowledgments and Disclosure of Funding

This work has been supported by the German Research Foundation through the Cluster of Excellence "Machine Learning – New Perspectives for Science" (EXC 2064/1:390727645) and the BMBF Tübingen AI Center (FKZ: 01IS18039A). The authors thank the International Max Planck Research School for Intelligent Systems (IMPRS-IS) for supporting Leena C Vankadara and Luca Rendsburg.

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
