# OpenReview forum: "Interpolation and Regularization for Causal Learning"
_NeurIPS.cc/2022/Conference — NeurIPS 2022 Accept_

### Official Review · Reviewer_XX7K · 2022-07-08

**Rating:** 6
**Confidence:** 1
**Soundness:** 3 good
**Presentation:** 3 good
**Contribution:** 3 good

**Summary:**

The paper gives a theoretical analysis about the relationship between interpolation and causal generalization. The paper proposes a new measurement of confounding strength \zeta, and discusses under different cases of \zeta, how the optimal strength of the regularization varies.

**Questions:**

N/A

**Ethics Review Area:**

["I don’t know"]

**Strengths And Weaknesses:**

I am not expert of the causal learning and theoretical statistical learning. My evaluation is only based on a general view of the paper, not diving into the details (proof, etc).
Strength:
1. the problem seems important and open
2. the theoretical results see solid and very interesting

Weakness:
1. This is a pure theory paper. While it is unnecessary, demonstrating the results on some real-world datasets/applications will be  promising and even more interesting.

---

> ### Author Response · Authors · 2022-08-01
> **Initial Author Response**
>
> Thank you for taking the time to read through our paper. We are glad that you found our results very interesting.
>
> **"Demonstrating the results on some real-world datasets/applications will be promising and even more interesting''**
> We theoretically characterize causal learning from observational data under a linear Gaussian model.
> We cannot claim that our results and qualitative statements generalize for datasets beyond a linear model. However, we believe that our limiting expressions (and the qualitative messages derived henceforth) would be rather robust to violations of certain model assumptions in the linear setting. For instance, due to the Universality phenomenon in the high-dimensional limit, we believe that our results can closely predict the behavior of causal risk under shifts in the distributions of the covariates as long as moments of order $(4 + \delta)$ for some $\delta >0$ are bounded.
> We conducted experiments to verify this claim and the corresponding results can be found here [https://bit.ly/3bsa7MJ](https://bit.ly/3bsa7MJ). These experiments compare our theoretically derived asymptotic risks with finite-sample risks of the min-norm interpolator and causally optimally regularized ridge regressor. Instead of Gaussian confounders $z\sim\mathcal{N}(0, I_l)$, we only fix the first two moments $0$ and $I_l$ and generate $z$ from (i) a heavy-tailed multivariate $t$-distribution with different degrees of freedom, and (ii) a finite mixture of Gaussians. Our experiments show that, for distributions with finite 4th moments, the finite-sample risks closely match the theoretical results. A detailed description of the parameters used for the experiments is described below the figures.
>
> For experiments on confounding strength estimation for real-world datasets, we refer the reviewer to the Experiments sections in [18,21].

---

> > ### Comment · Reviewer_XX7K · 2022-08-09
> > **Thanks for your response**
> >
> > Thanks for your response and additional experiments. I will raise my score.

---

> ### Comment · Area_Chair_QPao · 2022-08-08
> **please acknowledge the authors' response**
>
> Please acknowledge the authors' response.

---

### Official Review · Reviewer_hNPk · 2022-07-10

**Rating:** 6
**Confidence:** 3
**Soundness:** 4 excellent
**Presentation:** 4 excellent
**Contribution:** 4 excellent

**Summary:**

The causal risk (under do operator) of ridge regression estimator and min-norm interpolator have been studied. Intriguing and deep relationship between causal signal strength and regularization for ridge regression is studied.

**Questions:**

(1) It seems $\||\Gamma\||^2_{\Sigma}$ in line 122 on p.3 should’ve been $\||\Gamma\||_{\Sigma^+}^2$ based on Proposition A.1, in which the variance is $I-A^\top (A A^\top)^+ A$?

(2) Should the variance $y|do(x=x_0)$ be $\sigma_{\tilde \varepsilon}^2+\||\Gamma\||^2_{\Sigma^+}$ in line 132, 154, 155 on p.4, because $do(x=x_0)$ eliminates the dependence of $z$ on $x$ so $z^\top\alpha$ is also regarded as an error term?

(3) Perhaps it would be great to add something to clarify that the $E_{Y|X}$ in $R_X^C(\hat\beta)=E_{Y|X}[R^C(\hat\beta)]$ is taken with respect to $\hat\beta$ in line 214 on p.6.

(4) For line 227 and 228 in Theorem 3.1, to extend the results for ridge regression in Eq. (7) and (8) to min-norm estimator, I think one also needs $d>n$ or $d/n\to\gamma\in(1,\infty)$ in addition to $\lambda\to 0$?


**Limitations:**

None.

**Strengths And Weaknesses:**

+ The study of causal risk for linear interpolator is timely and impactful
+ Theoretical analysis is solid

---

> ### Author Response · Authors · 2022-08-01
> **Initial Author Response**
>
> Thank you for carefully reading the technical parts of our paper and for providing detailed comments. We addressed all the questions below and we hope that you would consider improving the overall score of the paper.
>
> **(1) Should $\lVert\Gamma\rVert_\Sigma^2$ (line 122,p.3) be $\lVert\Gamma\rVert_{\Sigma^+}^2$?**
> The term $\lVert\Gamma\rVert_\Sigma^2$ is correct. Following the proof of Proposition 2.1 in Appendix A, we arrive at the variance of $z^T\alpha|x$ which reads $\lVert\alpha\rVert^2 - \alpha^TM^T(MM^T)^+M\alpha$. The latter summand is not $\lVert\Gamma\rVert_{\Sigma^+}^2$, because $\Gamma\neq M\alpha$. Instead, we have $\Gamma=M^{-T}\alpha$, so we first use the identity $M^T(MM^T)^+M=M^+MM^TM^{-T}$ to arrive at $\lVert\Gamma\rVert_\Sigma^2$.
>
> **(2) Should the variance of $y|do(x=x_0)$ be $\sigma_{\tilde{\varepsilon}}^2+\lVert\Gamma\rVert_{\Sigma^+}^2$?**
> Aside from the fact that we have the term $\lVert\Gamma\rVert_{\Sigma}^2$ instead of $\lVert\Gamma\rVert_{\Sigma^+}^2$ as discussed above, it is true that the variance of $y|do(x=x_0)$ is given by $\sigma_{\tilde{\varepsilon}}^2+\lVert\Gamma\rVert_\Sigma^2$ instead of $\sigma_{\tilde{\varepsilon}}^2-\lVert\Gamma\rVert_\Sigma^2$. Thanks for noticing this typo! We have corrected the corresponding parts.
>
> **(3) Add clarification that the expectation on line 214 is taken with respect to $\hat{\beta}$.**
> The expectation is taken with respect to $Y|X$ as written in the paper, but we have added a clarification that $\hat{\beta}=\hat{\beta}(X,Y)$ is a function of these random variables.
>
> **(4) Does the $\lambda\to 0$ part in Theorem 3.1 require $\gamma>1$?**
> We can exchange the limits even for $\gamma \in (0, \infty)$. Deriving the risk of the min-norm estimator involves deriving the limiting expressions for bias and variance as for any $\lambda > 0$ and exchanging the limits $({n, p \rightarrow \infty})$ and $({\lambda \rightarrow 0^+})$ by an application of Arzela-Ascoli Theorem to show the existence of limits and then an application of Moore Osgood theorem to exchange the limits.

---

> > ### Comment · Reviewer_hNPk · 2022-08-09
> > **Thanks for your clarification**
> >
> > I have no further comments.

---

> > > ### Author Response · Authors · 2022-08-09
> > > **Clarification of score/Further concerns?**
> > >
> > > Since there are no further comments or concerns, we hope that the reviewer would consider improving their overall score for the paper. Especially since they acknowledge the soundness, contribution, and presentation with the highest scores. Alternatively, could the reviewer kindly clarify their overall score? If there are any additional concerns, we would also be happy to address them.

---

> ### Comment · Area_Chair_QPao · 2022-08-08
> **please acknowledge the authors' response**
>
> Please acknowledge the authors' response.

---

### Official Review · Reviewer_ryke · 2022-07-13

**Rating:** 7
**Confidence:** 4
**Soundness:** 4 excellent
**Presentation:** 3 good
**Contribution:** 3 good

**Summary:**

This paper investigates the risk for a (linear causal) parameter estimator in terms of interpolation and regularization. The authors propose several notions to study the difference between causal and statistical learning behavior and showed that causal risk curve exhibits a double-descent phenomenon. Further authors proved that, under a certain condition, a causal risk can be decreased with a stronger regularization than usual for learning statistical parameters.


**Questions:**

My only question is whether learning a statistical parameter with regularization (or through interpolation in overparameter regime) can be referred to as causal learning.

Thanks for an intriguing paper.

**Ethics Review Area:**

["I don’t know"]

**Limitations:**

- I would like to see how confounding measure can be estimated, which is then one can practically learn causal parameter. This seems wrong...


**Strengths And Weaknesses:**


Strengths
- Highly interesting questions and mathematically rigorous analysis.
- Concepts introduced and theoretical results can be beneficial to further study “causal learning” of parameters in the sense of modern “statistical learning”.


Weaknesses
- Regardless of (strong/weak) regularization, it is hard to accept (in my causal mind) that a regularization “enables” causal learning. Nudging parameters toward a little bit doesn’t seem qualified for the term “causal learning” (e.g., figure 2 is drawn without omega square = squared norm of confounding strength)
- Whether to use strong regularization or not depends on confounding strength measure (and SNR_S), where confounding strength measure would not be typically measured. (I didn’t check out [20, 21] where the authors claimed that they can estimate the confounding strength.) If Gamma can be measured, empirical validation would be desired.

Sorry if there is any misunderstanding. I will correct if the authors can point out any.

---

> ### Author Response · Authors · 2022-08-01
> **Initial Author Response**
>
> Thank you for your time and valuable feedback. We are pleased that you found our work intriguing.
>
> **It is hard to accept that regularization "enables'' causal learning. Can learning a statistical parameter with regularization be referred to as causal learning?**
>
> Thanks a lot for this question. This will help us strengthen this paper, especially its motivation and presentation. We will include this discussion in the main paper for the final version. To clarify what we mean by "causal learning'', let us consider the problem of statistical learning in high-dimensions (say $\gamma > 1$). In this setting, without additional assumptions, it is information-theoretically impossible to achieve statistical Bayes risk. However, we can still learn an estimator up to this lower bound. Another natural baseline for linear regression (analogous to better than random guessing for classification) would be the mean estimator (or the null estimator in our setting). We say that statistical learning is possible if we can achieve a lower risk than the null estimator which can be achieved in this setting by a ridge regression estimator corresponding to the optimal statistical regularization.
>
> Analogously, by "causal learning'' we refer to predictors which perform better than the null estimator, not necessarily only predictors which achieve Causal Bayes risk. This figure [https://bit.ly/3OL6ijj](https://bit.ly/3OL6ijj)
>  illustrates how much can be learned about the causal parameter $\beta$ in our setting of ridge regression on observational data as measured by our causal risk $\lVert\hat{\beta}-\beta\rVert_\Sigma^2$ if the confounding strength can be estimated.
>
> For each causal model, we compare the causal risk of the optimally regularized ridge-regressor $\hat{\beta}\_{\lambda_C^\ast}$ to two baselines, the ridge-regressor under statistical regularization $\hat{\beta}_{\lambda^\ast_S}$ and the null predictor $\hat{\beta}=0$.
> For $\zeta=0$, causal and statistical models coincide and both statistical and causal regularization achieve Bayes risk. As $\zeta$ increases, we see the advantage of causal regularization over the baselines (hence causal learning), although we do not achieve Bayes risk anymore. For large $\zeta$, causal regularization achieves Bayes risk again, because here $\beta\approx 0$, which is achieved by $\lambda_C^\ast=\infty$.
>
> Finally, we would also like to add here that understanding information-theoretic lower bounds for causal learning even in the linear setting is a fantastic open problem which we will mention in the final version.
>
> **Figure 2 is drawn without the squared norm of the confounding parameter $\lVert\Gamma\rVert^2=\omega^2$.**
> Dropping the constant $\tilde{\sigma}^2+\omega^2$ in Figure 2 is not meant to disguise the dependence of the causal risk on the confounding; we dropped it because $\tilde{\sigma}^2+\omega^2$ is the *causal Bayes risk* (see Proposition 2.1) and therefore even the true causal parameter $\beta$ incurs this causal risk of $\tilde{\sigma}^2+\omega^2$.
>
> **Would like to see how confounding strength can be estimated. If $\Gamma$ can be measured, empirical validation would be desired.**
> We agree with you that confounding strength cannot be estimated without any assumptions. The estimation procedure of [21] is valid only under the ICM assumption that we also discuss in our paper (that is, $\alpha$ and $\beta$ are drawn independently from rotationally invariant distributions). In short, under strong confounding, the statistical parameter $\tilde{\beta}=\beta+M^{-T}\alpha$ tends to be more aligned with small eigendirections of the covariance $\Sigma=MM^T$. This information is leveraged to estimate the confounding strength parameter.
>
> Empirically validating the confounding strength estimation procedure is beyond the scope of our paper and we refer the reviewer to the Experiments sections in [18,21]. Here, we rather focus on characterizing causal learning from observational data for a given confounding strength. That being said, we are currently working on extending the estimation procedure in [21] to the high-dimensional setting.

---

> > ### Comment · Reviewer_ryke · 2022-08-08
> > **.**
> >
> > Thanks for the detailed reply especially the answer to the first question, which clears up some of my misunderstanding of the core concept.

---

### Official Review · Reviewer_ST5R · 2022-07-13

**Rating:** 6
**Confidence:** 3
**Soundness:** 4 excellent
**Presentation:** 3 good
**Contribution:** 3 good

**Summary:**

This paper studies the problem of learning causal effects with the existence of hidden confounders under a simple linear causal model. The authors provide an analysis of this learning problem in observational settings from the perspective of interpolation and regularization. They drive precise asymptotic risk of the min-norm interpolator and the ridge-regularized regressors. They further propose a new measure of confounding strength to characterise rich behaviours of causal learning under the ICM principle.

**Questions:**

+ In the simple linear causal model, why is the dimension of $x$ assumed to be smaller than that of $z$?

+ If the variables considered in the paper are non-Gaussian, will it change the theoretical results? If so, how?

+ Throughout the paper, only the hard intervention is taken into account. What will happen to the soft intervention?

+ How will these theoretical results change in the nonlinear setting? As we know, under the nonlinear causal model, it is infeasible to represent the causal effect using only one parameter (e.g., $\beta$). In this case, I can imagine the theoretical results would be quite different.

**Limitations:**

The authors discussed the limitations but not the potential negative societal impact of their work.

**Strengths And Weaknesses:**

Strengths:
+ The problem studied in this paper is of importance to the community.
+ convincing theoretical analysis and easy to follow

Weaknesses:
+ The presentation of some paragraphs should be improved, since they involve a few unexplained terminology (e.g., U-shaped generalization, double-descent phenomenon, benign causal overfitting, etc), which is unfriendly to those unfamiliar with learning theory. I suggest that the authors add a bit explanations to them to improve readability.

+ The work only focuses on the simple linear causal models with Gaussian variables. Hence, their theoretical results are also limited to the linear setting. However, this is not the case in practice.

+ Empirical evaluation is limiting.

---

> ### Author Response · Authors · 2022-08-01
> **Initial Author Response (1/2)**
>
> We appreciate the reviewer’s time and effort in reviewing our manuscript. Below, we address all the questions in your review. We hope this persuades you to update the overall rating.
>
> **Extensions to non-linear settings.**
> You asked us how the results could be extended to non-linear settings. Thank you for this important question. We are already working on extending the analysis to structural causal models that arise in a reproducing kernel Hilbert space (RKHS) corresponding to a positive definite kernel (i.e, where the best statistical model $\tilde{f}$ and the best causal model $f$ are functions in some RKHS). There are two major technical challenges to deriving the theoretical analysis in such non-linear settings. Both are beyond the scope of this paper and are left for future work, but we will add a discussion of these extensions in the final version.
>
> - **Extend the definition of confounding strength $\zeta$ beyond the linear setting.** Since such a definition is non-trivial already in the linear setting, it is challenging to meaningfully generalize this to the non-linear setting. However, under non-linear causal models in the RKHS, we can naturally extend the definition of confounding strength by replacing the Euclidean norms with functional norms in the RKHS. Generalizing the analysis beyond this setting would require further careful consideration.
> - **Derive limiting expressions for the causal risk of regularized regressor from a hypothesis class of non-linear models.** In the case of kernel regression, this would be feasible via some recent random matrix theory results [27]. By optimizing the limiting expressions with respect to the regularization parameter, one can obtain the parameter that achieves the optimal causal risk and subsequently can identify the relationship between optimal causal regularization and confounding strength.
>
> In your review, you stated that it is ``infeasible to represent the causal effect using only one parameter $\beta$ under non-linear causal models''.
> We are not sure what the statement refers to here. Could you kindly clarify? In non-linear models, instead of learning a linear function parameterized by $\beta$, we would need to learn a function $f^*$ representing the structural function between the covariates and target variable.
>
> **Extensions beyond Gaussian distributions.**
>
> You also asked us if we can extend the analysis beyond the Gaussian setting. Thank you for another important question that helps us address the applicability of our results beyond our model setting. The analysis can be extended beyond the Gaussian setting by considering random variables generated by finite mixtures of Gaussians. The analysis can get considerably more technical and is left as future work, but we will add this discussion to the paper. Due to the Universality phenomenon in the high-dimensional limit, we believe that our limiting expressions (and the qualitative messages derived henceforth) would be rather robust to shifts in the marginal distribution as long as moments of order $(4 + \delta)$ for some $\delta >0$ are bounded. We conducted experiments to verify this claim and the corresponding results can be found here [https://bit.ly/3bsa7MJ](https://bit.ly/3bsa7MJ). These experiments compare our theoretically derived asymptotic risks with finite-sample risks of the min-norm interpolator and causally optimally regularized ridge regressor. Instead of Gaussian confounders $z\sim\mathcal{N}(0, I_l)$, we only fix the first two moments $0$ and $I_l$ and generate $z$ from (i) a heavy-tailed multivariate $t$-distribution with different degrees of freedom, and (ii) a finite mixture of Gaussians. Our experiments show that, for distributions with finite 4th moments, finite-sample risks closely match the theoretical results. A detailed description of the parameters used for the experiments is described below the figures in [https://bit.ly/3bsa7MJ](https://bit.ly/3bsa7MJ).

---

> > ### Author Response · Authors · 2022-08-01
> > **Initial Author Response (2/2)**
> >
> > **Extension to soft interventions.**
> > This is certainly an interesting question. The precise answer would depend on the type of soft interventions. Consider, for instance, the class of relative interventions where the structural dependence of the covariates $x$ is not destroyed as in the case of hard interventions but the observed covariates are merely perturbed (i.e., interventions of the form $do(x:=x+x^*)$). Then, it turns out that $\textrm{Causal risk}\_{\textrm{soft}} = \textrm{Causal risk}\_{\textrm{hard}} + \textrm{Statistical risk}$. From our results, it then follows that under ICM, $\lambda^{\textrm{stat}} \leq \lambda^{\textrm{causal}}\_{\textrm{soft}} \leq  \lambda^{\textrm{causal}}\_{\textrm{hard}}$. This also supports our intuition since under soft interventions, we typically aim to achieve a tradeoff between statistical and causal predictability. We will add this result to the main paper in the final version.
> >
> > **Missing discussion of potential negative societal impact.**
> > We thank the reviewer for this question. Causal inference has positive applications, for example, in the medical and scientific domains but one cannot rule out potentially negative applications. Our results address fundamental questions that clarify what is possible in general. Also, our results should not be taken as a justification for ignoring the complexity and hardness of causal learning in general. Learning causal relationships between two variables purely from observational data requires strong assumptions or additional information. Our results show that regularization can enable causal learning from observational data when one *can estimate confounding strength*. However, even when confounding strength is known, regularization alone does not achieve *causal Bayes risk* even asymptotically. This figure [https://bit.ly/3OL6ijj](https://bit.ly/3OL6ijj) illustrates how much can be learned about the causal parameter $\beta$ in our setting of ridge regression on observational data as measured by our causal risk $\lVert\hat{\beta}-\beta\rVert_\Sigma^2$. We will emphasize this in our Discussion.
> >
> > **Improve presentation of some paragraphs.**
> > We appreciate your comment on providing additional background on learning-theoretic terms such as U-shaped generalization and double-descent phenomenon to improve readability. We will add short explanations for these terms in the main paper and provide further background in the supplement.
> >
> > **Empirical evaluation is limiting.**
> > We theoretically characterize causal learning from observational data under a linear Gaussian model.
> > We cannot claim that our results and qualitative statements generalize for datasets beyond a linear model. However, we believe that our limiting expressions (and the qualitative messages derived henceforth) would be rather robust to violations of certain model assumptions in the linear setting. For instance, due to the Universality phenomenon in the high-dimensional limit, we believe that our results can closely predict the behavior of causal risk under shifts in the distributions of the covariates as long as moments of order $(4 + \delta)$ for some $\delta >0$ are bounded.
> > We conducted experiments to verify this claim and the corresponding results can be found here [https://bit.ly/3bsa7MJ](https://bit.ly/3bsa7MJ). These experiments compare our theoretically derived asymptotic risks with finite-sample risks of the min-norm interpolator and causally optimally regularized ridge regressor. Instead of Gaussian confounders $z\sim\mathcal{N}(0, I_l)$, we only fix the first two moments $0$ and $I_l$ and generate $z$ from (i) a heavy-tailed multivariate $t$-distribution with different degrees of freedom, and (ii) a finite mixture of Gaussians. Our experiments show that, for distributions with finite 4th moments, the finite-sample risks closely match the theoretical results. A detailed description of the parameters used for the experiments is described below the figures.
> >
> > For experiments on confounding strength estimation for real-world datasets, we refer the reviewer to the Experiments sections in [18,21].
> >
> > **Why assume dimension $d$ of $x$ is smaller than the dimension $l$ of $z$?**
> > We consider the setting $d\leq l$ since it is more general. The case $l<d$ would lead to rank restrictions of the covariance
> > of $x$ because $\text{Cov}(x)=MM^T$ with $M\in \mathbb{R}^{d\times l}$. Additionally, the case $l<d$ is contained in $l\geq d$ in the sense that we can simply restrict the hyperparameters $M$ and $\alpha$. Specifically, we could choose $M=(M^\prime, 0)$ and $\alpha=(\alpha^\prime, 0)$ with $M^\prime\in \mathbb{R}^{d\times l^\prime}$ and $\alpha^\prime\in \mathbb{R}^{l^\prime}$ for some $l^\prime<l$.

---

> ### Comment · Area_Chair_QPao · 2022-08-08
> **please acknowledge the authors' response**
>
> Please acknowledge the authors' response.

---

### Meta-Review · Area_Chair_QPao · 2022-08-23

**Recommendation:** Accept
**Confidence:** Certain

**Metareview:**


The consensus view was that the theoretical analysis was important, fundamentally novel and very interesting.  The paper was well written.

**Award:**

No

---

### Decision · Program_Chairs · 2022-09-14

Accept